



# Observational data from uncrewed systems over Southern Great Plains

Fan Mei[1], Mikhail S. Pekour[1], Darielle Dexheimer[2], Gijs de Boer[3,4,5], RaeAnn Cook[2], Jason Tomlinson[1], Beat Schmid[1], Lexie A. Goldberger[1], Rob Newsom[1], Jerome D. Fast[1]

[1]Pacific Northwest National Laboratory, Richland, WA, 99352, USA
[2]Sandia National Laboratories, Albuquerque, NM, 87185, USA
[3]Cooperative Institute for Research in Environmental Sciences, University of Colorado, Boulder, CO, 80309, USA
[4] NOAA Physical Sciences Laboratory, Boulder, CO, 80305, USA
[5] Integrated Remote and In Situ Sensing, University of Colorado, Boulder, CO, 80309, USA

*Correspondence to*: Fan Mei (fan.mei@pnnl.gov)

**Abstract.** Uncrewed Systems (UxS), including uncrewed aerial systems (UAS) and tethered balloon/kite systems (TBS), are significantly expanding observational capabilities in atmospheric science. Rapid adaptation of these platforms and the advancement of miniaturized instruments have resulted in an expanding number of data sets captured under various environmental conditions by the Department of Energy (DOE) Atmospheric Radiation Measurement (ARM) user facility. In 2021, observational data collected using ARM UxS platforms, including seven TigerShark UAS flights and 133 tethered balloon system (TBS) flights, were archived by the ARM Data Center (https://adc.arm.gov/discovery/#/) and made publicly available at no cost for all registered users (https://www.doi.org/10.5439/1846798) (Mei and Dexheimer, 2022). Note that a specific directory has been created for the anonymous reviewer to access the data at https://adc.arm.gov/essd/. These data streams provide new perspectives on spatial variability of atmospheric and surface parameters, helping to address critical science questions in Earth system science research. This manuscript describes the DOE UAS/TBS datasets, including information on the acquisition, collection, and quality control processes, and highlights the potential scientific contributions using UAS and TBS platforms.

## 1. Introduction

Expanding development and deployment of various uncrewed aircraft systems (UAS) result in increasing opportunities for these platforms to provide high-quality atmospheric measurements (Stephens et al., 2000; Hobbs et al., 2002; Villa et al., 2016; De Boer et al., 2020b). Several recent atmospheric science campaigns have provided perspectives on the planetary boundary layer with both UAS (Fladeland et al., 2011; Villa et al., 2016; Adkins and Sescu, 2018; Barbieri et al., 2019; Chen et al., 2020; De Boer et al., 2020b; De Boer et al., 2020a) and tethered balloon system (TBS) (De Boer et al., 2018; Creamean et al., 2021). These studies have taken advantage of various scales of UAS platforms, ranging from very small (de Boer et al., 2019) to very large (Intrieri et al., 2014). The use of mid-sized UAS (Group 3, weight more than 55 lbs, but less than 1320 lbs) has



also gained traction in support of atmospheric science due to the ability of such platforms to carry out remotely-sensed and in situ observing missions with collocated instruments in a large payload (>25 lbs) (Bates et al., 2013; Solbø and Storvold, 2013; Boer et al., 2016; Barfuss et al., 2018; Reineman et al., 2018; Lee and Kim, 2019). Extended endurance (up to 12 hours) and

operating range (at altitudes from 0 to 9000 m) support opportunities for extensive data collection, especially when deploying simultaneously with other ground and mobile observation stations (Stephens et al., 2000; Lawson et al., 2011; Bates et al., 2013; Boer et al., 2016; Barfuss et al., 2018; De Boer et al., 2018; De Boer et al., 2019a; De Boer et al., 2019b; Vihma et al., 2019).

In 1989, the US Department of Energy (DOE) established the Atmospheric Radiation Measurement (ARM) user facility. A

series of instrumented platforms were developed to provide comprehensive measurements to support climate research, providing long-term records of the atmospheric state over a wide range of conditions (Mather and Voyles, 2013). The ARM-Unmanned Areal Vehicle (UAV) program was first introduced in 1991 and demonstrated how measurements from UAV platforms contribute to our understanding of cloud and radiative processes through eight flight campaigns and over 140 hr. of science flights with three different UAV platforms (Stephens et al., 2000). With the continuing maturation of ARM TBS/UAS

capabilities, the DOE ARM facility restarted UxS observing around 2013, supporting various TBS and UAS deployments within restricted airspace over Oliktok Point, Alaska (De Boer et al., 2016; De Boer et al., 2018; De Boer et al., 2019b; Creamean et al., 2021). After re-engaging with such activities, the DOE ARM facility procured and instrumented a mid-size UAS intending to provide new sampling capabilities globally (De Boer et al., 2018). Simultaneously, ARM engaged the community to articulate scientific needs (Mei et al., 2020a), guided and expanded approval requirements for the operation of

UAS and TBS (Bendure et al., 2019), supported the evaluation of numerous UAS platforms at ARM user facilities (Jacob et al., 2018; Schuyler et al., 2019), and practiced joint UAS-Balloon activities (Dexheimer et al., 2018).

This development path has resulted in an improved ability of the DOE ARM program to support lower-atmospheric research. One specific example of the type of research that is being undertaken involves detailed observations of the atmospheric boundary layer (ABL). Accurate prediction of ABL processes is challenging due to the diurnal cycle of heating and cooling of

the various land surface (Banta, 2008; Friedrich et al., 2012; Klein et al., 2014). Known as a "hot spot" for land-atmosphere interactions, the ARM Southern Great Plains (SGP) atmospheric observatory is located in central Oklahoma, where significant diurnal variability is observed in both thermodynamic and kinematic properties (Klein et al., 2015). ARM observations from this location are leveraged to understand the occurrence and impact of mesoscale disturbances to the ABL and connections between ABL processes and the life cycle of shallow convective clouds (Fast et al., 2019). In this sense, extending the surface-

based data collected at the SGP facility to include coincident airborne observations is critical in understanding the multitude of processes driving the ABL and its impacts on cloud development, weather and climate. To meet this community need, the ARM user facility has implemented short-term routine TBS and UAS measurements at the SGP atmospheric observatory, specifically including enhanced observations from a mid-size UAS (Mei et al., 2019; Schmid et al., 2020).

In 2021, the ARM TBS team was routinely deployed at three sites within the SGP network for coordinated flights at two of

these three sites: the central facility (C1, Lamont, OK) (Mather and Voyles, 2013) and two extended facilities, EF9 and E36.





The TBS team carried out two-week missions in February, May, July, and October 2021, providing vertical profiles of atmospheric parameters, including aerosol properties. In addition, between Nov. 8 and 16, 2021, a mid-size UAS equipped with miniaturized scientific instruments operated over the SGP central facility, collecting an extended dataset. The following sections of this paper provide descriptions of the TBS and UAS operated at the SGP site and the instrumentation carried by

these platforms. Subsequent sections offer insight into the flight patterns used to collect data and data processing, and quality control; we conclude with a discussion of the data file structure and availability.

## 2. TBS and UAS description and flights

### 2.1. TBS description and flights

The DOE ARM program has explored the scientific potential of the TBS since 2013. The ARM TBS is an uncrewed system

(UxS) composed of a helium-filled balloon, tether, winch system and a suite of sensors, and currently, the ARM has two sets of them (Dexheimer, 2018). The ultimate goal is to daily deploy the TBS for two weeks (as weather permits) at the ARM observatory and spread four deployments throughout a year to capture the seasonal variation. Initial progress toward routine observations was made at Oliktok Point (OLI), AK, in conjunction with the deployment of the third ARM mobile facility (AMF3) in 2014 (De Boer et al., 2016; De Boer et al., 2018). In addition, flights within DOE managed special use airspaces

(R-2204 and W-220) at and north of OLI promoted operation of UASs and TBSs and supported the development of publicly available data streams through the ARM Data Archive (Dexheimer, 2018; De Boer et al., 2019a; De Boer et al., 2019b; Dexheimer et al., 2019; Creamean et al., 2021). As discussed above, 2020 saw an extension of routine TBS observations to the ARM Southern Great Plains (SGP) atmospheric observatory. Despite the COVID-19 pandemic, the TBS team managed to carry out four missions (133 flights) at the SGP observatory in 2021. Coordinated TBS flights deploying instruments outlined

in Table 1 occurred at two sites simultaneously, as shown in Fig. 1. These flights took place over the 160 acres of cattle pasture and wheat fields that make up the SGP, including the central facility C1 and extended facilities EF9 and E36. The three sites are stretched from north to south, with the central facility being in between the outer two. Together, these three sites are proposed to study the spatial variance of atmospheric properties such as the distribution of aerosols.



**Figure 1.** Map of the TBS deployment locations for flights conducted in 2021. (Dari and RaeAnn)

The TBSs were operated under the two (profiling and loitering) flight patterns shown in Fig. 2 and S1 when surface wind speeds were less than 10 m/s. During the profiling flight pattern, the TBS launched, carrying the payload (as shown in Fig. S2) to a maximum operating height (up to 2 km depending on the FAA approval), and then descended to the surface to complete one profile. The maximum payload weight varied between 10-25 kg, depending on surface wind speeds. The TBS employed a loitering flight pattern for targeted sampling. The system would ascend to a designated altitude, such as below the cloud base or inside a cloud, then remain static at that altitude for the desired period, i.e., 5 hours, before descending to the surface. During the 2021 SGP deployment, most of the flights were flown with the profiling pattern (more info in Table S3). The purpose of these flights was to collect seasonal observations of boundary layer atmospheric properties and aerosol vertical distributions that could be used to create altitude resolved statistics.



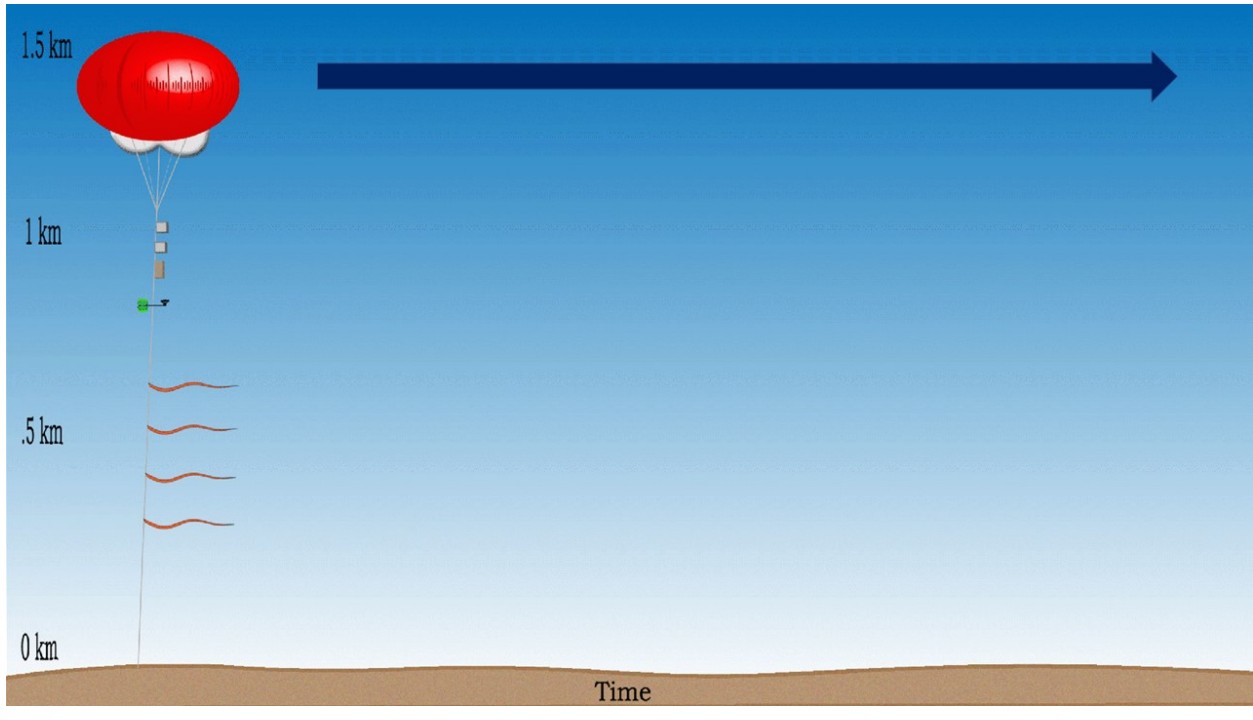

(a)

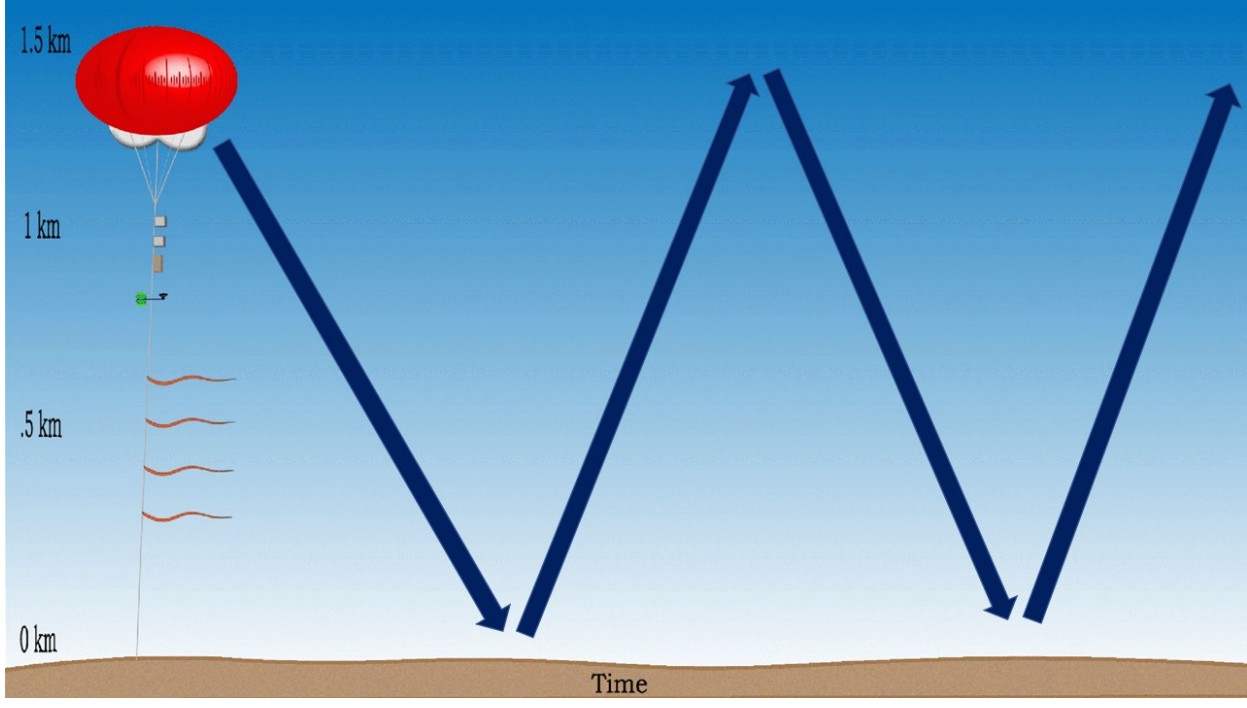

(b)



**Figure 2.** TBS typical flight pattern (a) loitering flight and the balloon is fixed at one altitude; (b) profiling flight and the balloon is going up and down. The arrows indicate the altitudes of the balloon remain during the flight.

Table 1. The ARM TBS instrumentation deployed during the 2021 SGP missions.

| Instrument | Property Measured | Status |
|---|---|---|
| iMet RSB-1 and RSB-4 radiosondes (multiple) | Pressure, Temperature, RH, 3D GPS | ARM Facility |
| iMet XQ2 UAV Sensor (multiple) | Pressure, Temperature, RH, 3D GPS | ARM Facility |
| Sensornet Oryx DTS<br>Siliax XT DTS | Distributed temperature sensing at 2 m and 0.5 m spatial resolution and 0.08 °C accuracy | SNL owned |
| 40C cup anemometers (8 units) | 1 Hz wind speed | ARM Facility |
| Printed Optical Particle Spectrometer (POPS, 6 units) | Aerosol size distribution from 140 nm to 3 μm | ARM Facility + SNL owned |
| Condensation Particle Counter (CPC) Model 3007 (4 units) | Total aerosol concentration from 0.01 μm to 1 μm | ARM Facility + SNL owned |
| TBS impactor (TBI, 6 units) | Size-resolved chemical composition at four cut-off sizes (0.25, 0.5, 1.0, 2.5 μm) | ARM/EMSL Facility |

The instruments deployed on the TBS for the routine measurements are listed in Table 1. Except for the TBI, the rest of the instruments were measured at a 1 Hz sampling rate. More details about the TBS instruments have been discussed previously in several publications (De Boer et al., 2018; Dexheimer, 2018; Dexheimer et al., 2019; Mei et al., 2020b; Creamean et al., 2021). More information about the TBS 2021 flights is listed in Table S3.

## 2.2. UAS description and flights

### 2.2.1. Mid-size UAS description

Located at the Pacific Northwest National Laboratory (PNNL), the ARM Aerial Facility (AAF) currently operates the Navmar Science Corporation (NASC) ArcticShark Uncrewed Aerial System (UAS) (De Boer et al., 2018), which is a custom-built variant (TS-B3-XP-AS) of the standard TigerShark Block 3 (TS-B3-XP) UAS, as detailed in Table S1. While the ArcticShark was undergoing upgrades and modifications in 2021, the AAF staff collaborated with the Mississippi State University (MSU) Raspet Flight Research Laboratory (RFRL) to integrate the ArcticShark scientific payload on RFRL TS-B3-XP TigerShark (Fig. 3). Between Nov. 2 and 19, 2021, both teams traveled to Blackwell-Tonkawa Municipal Airport (BKN) to conduct a series of flights with the RFRL TigerShark. The primary objective of this flight effort was to demonstrate the operational ability and durability of the payload (listed in Table 2) on an airframe similar to the ARM ArcticShark. In addition, these



flights (flight detail in Table S4) provided a significant amount of data that were used to evaluate data quality and provide an initial demonstration of the data to the atmospheric research community.

Table 2. The ARM UAS payload instruments deployed on the TigerShark during the SGP mission in 2021

| Instrument | Description | Source/Supplier |
| --- | --- | --- |
| VectorNav INS (VN-300) | Position and velocity | VINS |
| Aircraft Integrated Meteorological Measurement System - 30 (AIMMS-30) | 5-port air motion sensing: true airspeed, angle-of-attack, sideslip Meteorology: temperature and relative humidity. Inertial navigation system/global positioning system: position, velocity, attitude | Aventech |
| HS-2000DP | Temperature, relative humidity | Procon |
| Aerosol Isokinetic Inlet | Sample stream of dry aerosol, sizes < 5 microns | PNNL build |
| Mixing Condensation Particle Counter (MCPC) | Total aerosol concentration >0.007 µm | Brechtel Inc. |
| Miniaturized Optical Particle Counter (MOPC) | Aerosol size distributions 0.18 to 10 µm | Brechtel Inc. |
| Single Channel Tricolor Absorption Photometer (STAP) | Aerosol light absorption | Brechtel Inc. |
| Portable Optical Particle Spectrometer (POPS) | Aerosol size distribution 0.15 to 3 µm | Handix Scientific |
| Aerosol Filter Sampler | Offline aerosol chemical composition | Brechtel Inc. |
| Gas analyzer (Li-840A) | Concentration of $CO_2$ and $H_2O$ | LI-COR |
| Infrared radiometer | Surface temperature | Apogee |
| Multispectral Imager (Altum) | Multi-Spectral images | MicaSense |

All the above sensors acquired data at a 1 Hz sampling rate. In addition, the AIMMS-30 also recorded binary raw datasets at
a higher frequency (detailed in section 3.2.1), which can be reprocessed to produce higher time resolution (10 Hz) data. The primary aerosol suite includes five aerosol sensors: an advanced mixing condensation particle counter (MCPC), a miniaturized optical particle counter (mOPC), a single-channel tricolor absorption photometer (STAP), a time-resolved aerosol filter sampler, and a POPS (Mei, 2020; Mei and Goldberger, 2020a, b; Mei et al., 2020 ). Those five sensors were mounted on the top layer of the multiple instruments stackable tower (MIST), which is the central payload infrastructure (Mei et al., 2019).
This aerosol payload is routinely evaluated against the PNNL aerosol ground mobile station (AGMS, detailed in Table S2). As shown in Fig. 3, the aerosol total number concentration measured by the MCPC was compared with a butanol-based CPC (TSI, model 3772) and a water-based CPC (Aerosol Device, Magic CPC). In addition, the aerosol absorption coefficients were compared between the STAP and a PSAP (Particle Soot Absorption Photometer). The size-resolved aerosol number concentrations measured by MOPC and POPS were matched against a UHSAS (DMT, Ultra-High Sensitive Aerosol
Spectrometer) data in the overlapping size ranges.

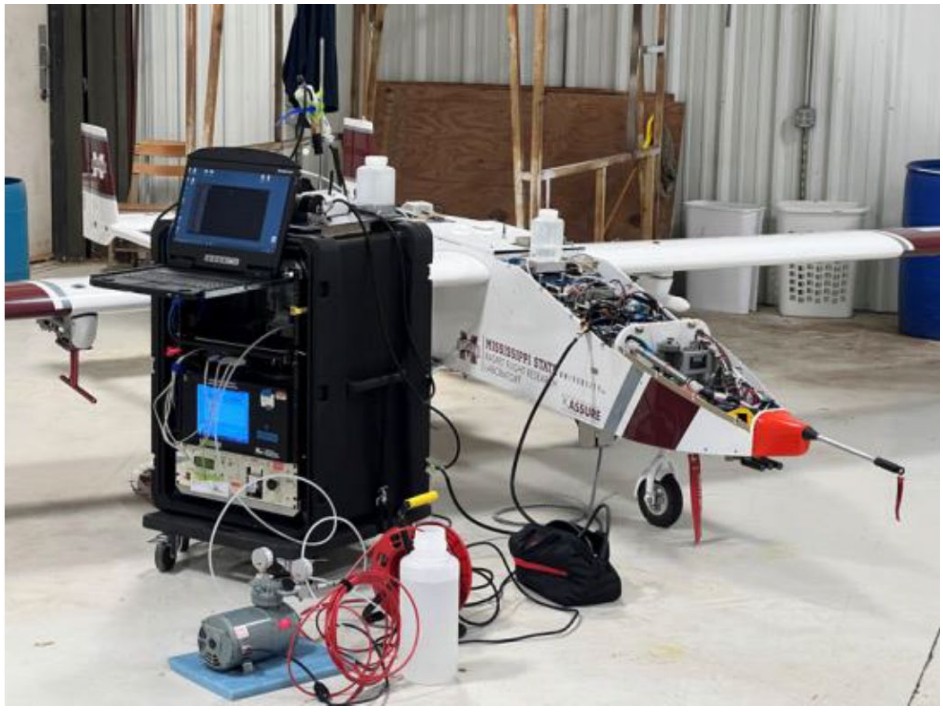

**Figure 3.** the AAF UAS payload sensors installed on the MSU TigerShark were compared with the PNNL aerosol ground mobile station (AGMS) before each flight.

In addition to the aerosol instrumentation, the AAF has capitalized on recent advances in instrument miniaturization to include

a variety of sensors on the TigerShark, as listed in Table 2. Together, these sensors observe atmospheric state (temperature, pressure, humidity, three-dimensional wind), gas concentrations ($H_2O$ and $CO_2$), surface temperature, multi-spectral surface images, and aerosol properties from the sensors described above (aerosol number concentration, size distribution, absorption coefficients, and chemical composition).

A forward-facing inlet was designed to ensure high-quality aerosol data (Fig. 3 and S3); the inlet provides isokinetic sampling

by maintaining the sample airspeed at the tip of the inlet equal to the true air speed.

### 2.2.2. UAS scientific flight patterns

Several meteorological applications benefit from collecting frequent and detailed profiles of temperature, relative humidity (RH), pressure and wind speed and direction. We have designed several flight patterns to match the requirements of specific sampling strategies (Fig.4). Using flight pattern A, we can survey the flight area using multiple spiral flight patterns during the

deployment. The accumulated data from spiral flight patterns (pattern A) informs flight planning around science target areas such as inversion layers, entrainment zones, and aerosol pollution layers. Flight patterns B and C are advantageous for profiling aerosol properties and turbulence parameters using vertically-distributed "level" flight legs that allow for the extended collection of statistics at a given altitude. Flight pattern D was targeted to surveying the surface properties over a larger area



from a single altitude. This "lawn mowing" flight pattern design ensures that the captured downward-looking multi-spectral
images overlap over 75% of the same surface area (projected on the ground) for post-processing.

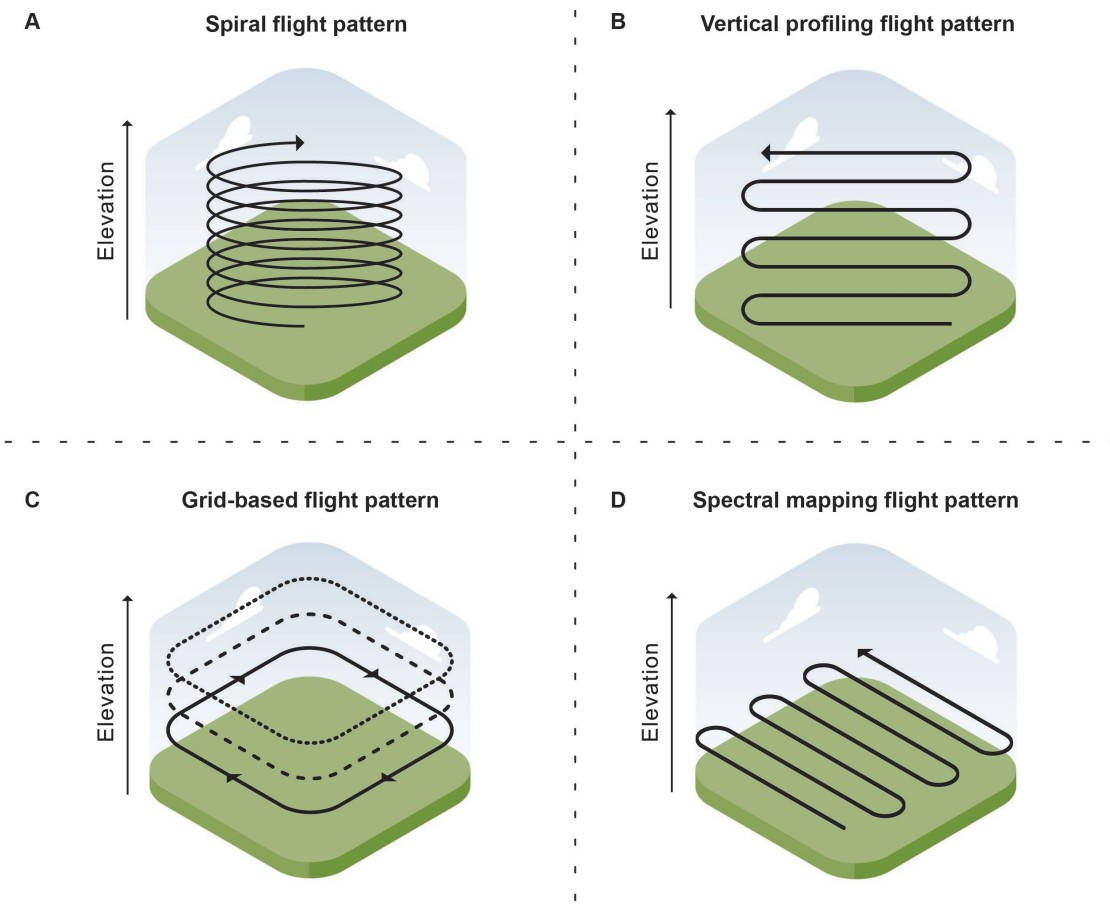

**Figure 4.** Four typical flight patterns used during the deployment (a) spiral pattern, (b) vertical profile ("Ladder") pattern, (c)
Grid-based ("Parking Garage") pattern, and (d) Spectral mapping ("Lawn Mowing") pattern

## 3.   Data quality control

### 3.1.   TBS data quality control

Platform position and meteorological data collected using the iMet were time-synchronized with the aerosol measurement
data, such as the POPS and CPC at a 1 Hz sampling rate. The initial time stamps of TBS flights were established based on the
launch and landing log. Per the manufacturer's specification, the iMet sensor has an accuracy of ±0.2 °C for ambient
temperature (range -90 to 50 °C), ±5 for the relative humidity (%, range 0-100%), and ±0.5 hPa for the ambient pressure (range
10-1200 hPa). The GPS altitude has ±15 m accuracy, with a position accuracy of ±10 m, and the wind velocity accuracy is





±0.1 m/s during loitering. A previous study compared the temperature measured by a fiber-optic distributed temperature sensing (DTS) system with the iMet sensor and concurrent radiosonde launches. The correlation $R^2$ between radiosonde and DTS measurements was about 0.99, and a root mean square error (RMSE) of 0.6 ℃. The mean RMSE between the iMet and DTS measurements improved from 0.39 ℃ to 0.32 ℃ with the correction derived from nine TBS flights (Dexheimer et al., 175    2019).

In 2021, the aerosol particle number concentrations were measured with a diffusion dryer installed upstream of the inlet line (RH < 40%). In addition, the aerosol particle size distribution was measured by POPS under ambient conditions (no dryer applied) in 2021. The calibrations for both CPC and POPS were performed before and after each deployment (Kuang and Mei, 2016; Mei et al., 2020b; Mei and Pekour, 2020; Bezantakos and Biskos, 2022), and all flow rates were periodically checked 180    in the field. The CPC sample flow rate is usually 100 $cm^3$/min, and the total flow rate is 700 $cm^3$/min with a variability of ±5%. The manufacturer specified the lower cut-off size $D_{50}$ =10 nm, where $D_{50}$ is defined as a particle diameter with 50% of the detected sample particles.

We periodically run all CPCs side-by-side during the deployment for comparison, as shown in Fig. 5. The total number concentration variation among all three units was below 15% at all times during the 2021 deployment period. Each POPS unit 185    was carefully evaluated in the controlled lab environment before each deployment by comparison to a scanning mobility particle sizer and an ultra-high sensitivity aerosol spectrometer to ensure the accuracy of sizing and counting (Mei et al., 2020b). When the UHSAS was operational (Cai et al., 2008) at the 3[rd] ARM mobile facility (AMF3 at Oliktok Point, AK, 2018) and the ARM SGP observatory (Aerosol Observation Station (AOS) at OK, 2019), we compared multiple POPS against the UHSAS on the ground. This comparison resulted in excellent agreement in the size distribution measurement, as shown in 190    Fig 6. When the total number concentration of POPS was over 4000 $cm^{-3}$, the coincidence error caused about 25% undercounting, and the POPS data would be flagged as questionable (Mei et al., 2020b). The laser temperature of the POPS was also monitored and is expected to stay between 25 and 65 ℃ to minimize the uncertainty in the size determination. Any out-of-range laser temperature will trigger a "bad" value flag in the data file (Mei et al., 2020b).



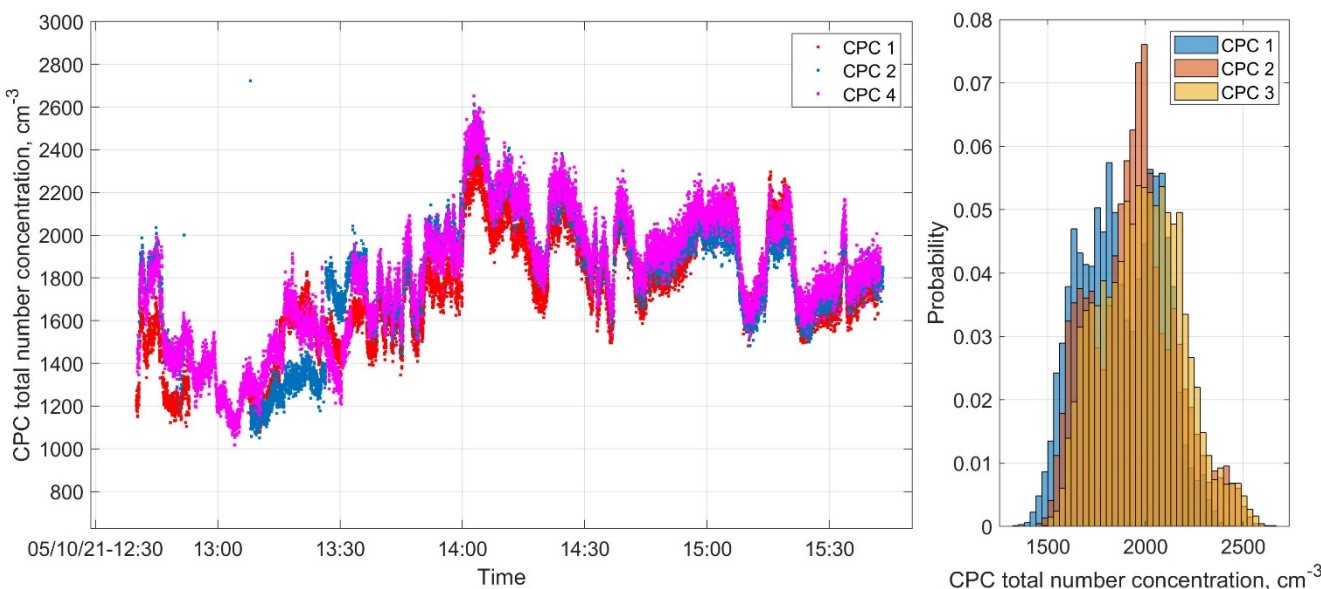

**Figure 5.** Aerosol total number concentration comparison between three CPCs at the top of a trailer. Left: time series of the total number concentrations from three CPCs. Right: Histogram of total number concentration for three CPCs during the comparison period 13:40 – 15:30, 05/10/221 (local time).

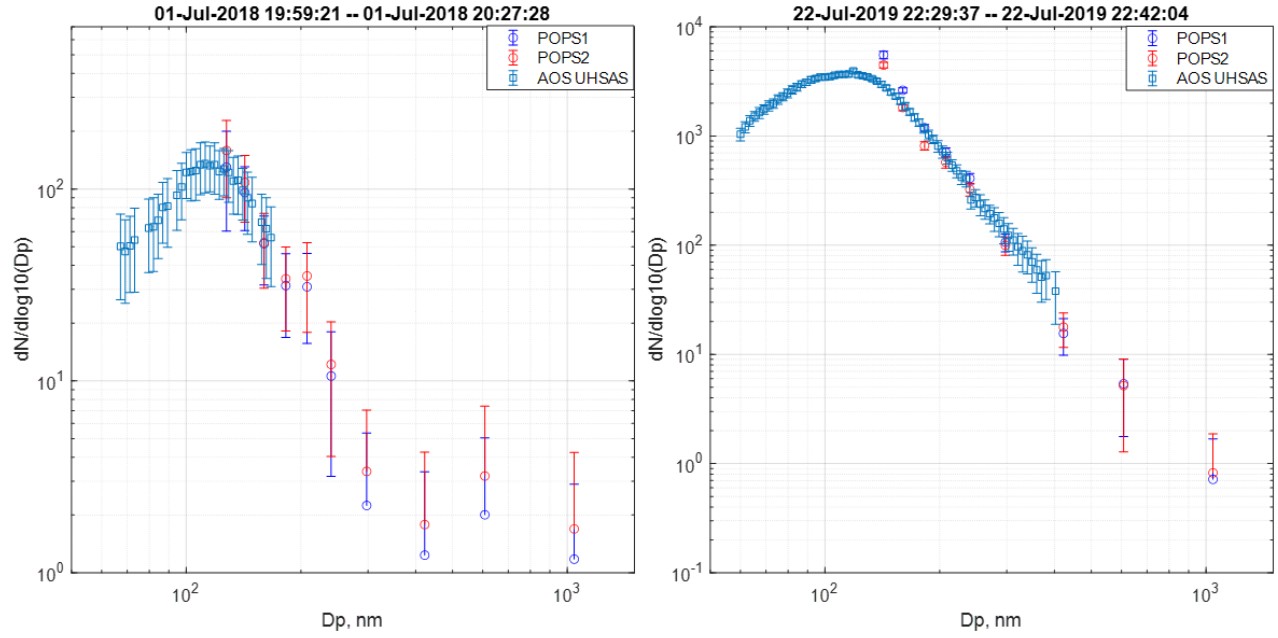

**Figure 6.** Aerosol size distribution comparison between TBS POPS and AMF3 UHSAS in 2018 and AOS UHSAS in 2019.



### 3.2. UAS data quality control

#### 3.2.1. Aircraft integrated meteorological measurement system data

The aircraft integrated meteorological measurement system (AIMMS-30, Aventech Research Inc.) provides accurate GPS position, platform velocity, aircraft attitude angles, and meteorological data, including wind vector

(https://aventech.com/products/aimms30.html). The manufacturer specifies that the accuracy of static pressure is 100 Pa (range: 0-1100 hPa), and the accuracy of pressure difference is 25 Pa (the pitot-static differential range: 0 -70 hPa; pressure difference ranges for the angle of attack (AOA) and the angle of sideslip (AOS): -70 - 70 hPa). The 3-axis accelerations have an accuracy of 0.002 g (range: -10 g – 10 g), and the 3-axis angular rates have an accuracy of 0.025 deg/s (range: -150 – 150 deg/s). The temperature sensor has an accuracy of 0.30 ℃ in the operating range of -40 to 50 ℃. The relative humidity sensor

has an accuracy of 2 % in the operating range of 0- 100 %. The precision of these measurements and an AIMMS-30 calibration flight are critical for accurately understanding the environmental condition during flight, especially to ensure the collection of a high-quality wind dataset. The measured true air speed (TAS) was validated by plotting the speed difference between the aircraft's ground speed and TAS against the aircraft heading angle when the UAS was flying in circles. The TAS errors derived for Aeroprobe, AIMMS-30, and PICCOLO (Pitot-static system) are 1.98, -0.57, and 2.53 m/s, as shown in Fig S4.

The AIMMS-30 calibration uses two different calibration procedures explicitly designed for the TigerShark platform, as shown in Fig.S5. The air data probe (ADP) associated with the AIMMS-30 was installed at 1.7 m from the center of the platform on the right wing, and the GPS antennas (referred to as the antenna baseline) were 6.54 m apart. First, the pilot and operator flew a level flight in the general direction of true north and south, manually between 27-41 m/s (53-80 kts), to perform the aerodynamic calibration maneuver to help determine aerodynamic errors induced by the aircraft itself. Then, the aircraft was

operated to complete six 360 degree orbits at three different bank angles (8, 13 and 17 degrees) to complete an inertial system calibration maneuver and capture minor alignment errors (i.e., cross-axis error) between the gyros, accelerometers, GPS antenna baseline and the primary reference frame of the inertial measurement unit (IMU). After the calibration flight, the updated setup parameters and calibration coefficients were programmed into the AIMMS-30 to ensure calibrated real-time wind data recording. During the operation, combining the measurements from the ADP, the GPS, and the IMU equipped with

the AIMMS-30 provides the wind speed accuracy of 0.5 m/s at three dimensions (north, east, vertical components) at 150 knots (77.2 m/s). The AIMMS wind parameters, such as wind speed, wind direction and wind components, were compared with the ARM Doppler Lidar (DL) retrieved wind parameters (Newsom et al., 2019) in Fig. 7. The ARM DL is an active remote-sensing instrument that provides time and range-resolved measurements (Newsom and Krishnamurthy, 2020). During the flight on 11/09/2021, the airplane flew a "ladder" pattern (pattern B) above the SGP central facility at different altitudes

(915 m, 1069 m, 1221 m, 1373 m, 1525 m, and 1610 m), while the DL operated in the near-infrared (IR: 1.5 microns) and provided accurate height-resolved measurements of wind speed and direction every 15 mins. We achieved a reasonably good comparison between the AIMMS and DL wind parameters. Further statistic studies and turbulence structure studies will be conducted with data from future deployments.



The ADP internal sampling rate is 200 Hz, and the IMU and GPS have an internal sampling rate of 250 Hz. The platform position, including latitude and longitude, was output by the AIMMS-30 at 1 Hz. The aircraft altitude has various inputs – GPS and pressure altitude. Due to drift throughout a flight, the pressure altitude potentially has more considerable uncertainty, which can be corrected using the known altitude at the airport location during post-processing. Furthermore, the GPS signal sufficiency limits the accuracy of the GPS altitude. Thus, we checked each flight's data and made sure to receive at least 6 GPS signals for each antenna. The temperature sensor has an accuracy of 0.3 °C, and the relative humidity (RH) accuracy is

2% as per manufacturer specification. We compared the dew point temperature derived from AIMMS-30 measured temperature and RH (Alduchov and Eskridge, 1996) with the dew point temperature measured by the $H_2O$ sensor (LiCor 840); the results are shown in Fig S6. The accuracies of the concentration measurements by LiCor 840 are better than 1.5% for both $CO_2$ and $H_2O$ measurements, based on the manufacturer's specifications.

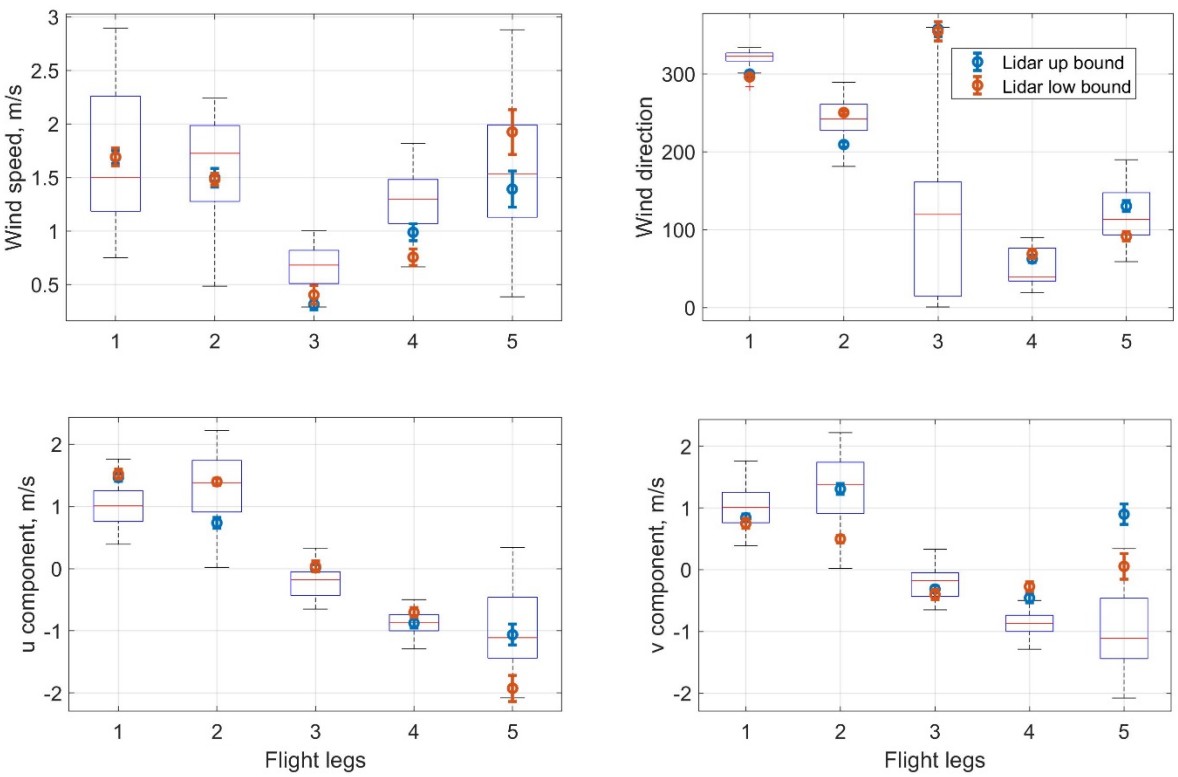

**Figure 7**. Wind parameters comparison between the AIMMS-30 and Lidar from five "leveled" flight legs at different altitudes on 11/09/2021. The average altitudes from flight legs 1 to 5 are 915 m, 1069 m, 1221 m, 1373 m, 1525 m, and 1610 m. On each box (representing the AIMMS data), the central mark indicates the median, and the bottom and top edges of the box indicate the 25th and 75th percentiles, respectively. The whiskers extend to the most extreme data points not considered outliers, and the outliers are plotted individually using the '+' marker symbol. The blue and red circles with the error bar are

250                                                         DL retrieved data.

### 3.2.2. Aerosol payload data

Aerosol measurements aboard the UAS platform were conducted in "dry" conditions because the aerosol inlet used for sampling was equipped with a nafion dryer ensuring relative humidity below 40% for the air sample. The aerosol inlet was designed by PNNL for a mid-size UAS platform (ARM technical report is under review). The inlet sampling condition was
checked with the help of a 5-port gust probe (Aeroprobe by AirData) that was mounted in place of the inlet tip for one flight. In Fig. 8, the AIMMS TAS is plotted as a blue bar in the top histogram, and the TAS measured at the tip of the isokinetic inlet is shown in orange. The median values from both measurements agree well with each other. The difference between the two TAS has a narrow distribution shown in the bottom histogram, which suggests the inlet was operated under the isokinetic condition as designed.

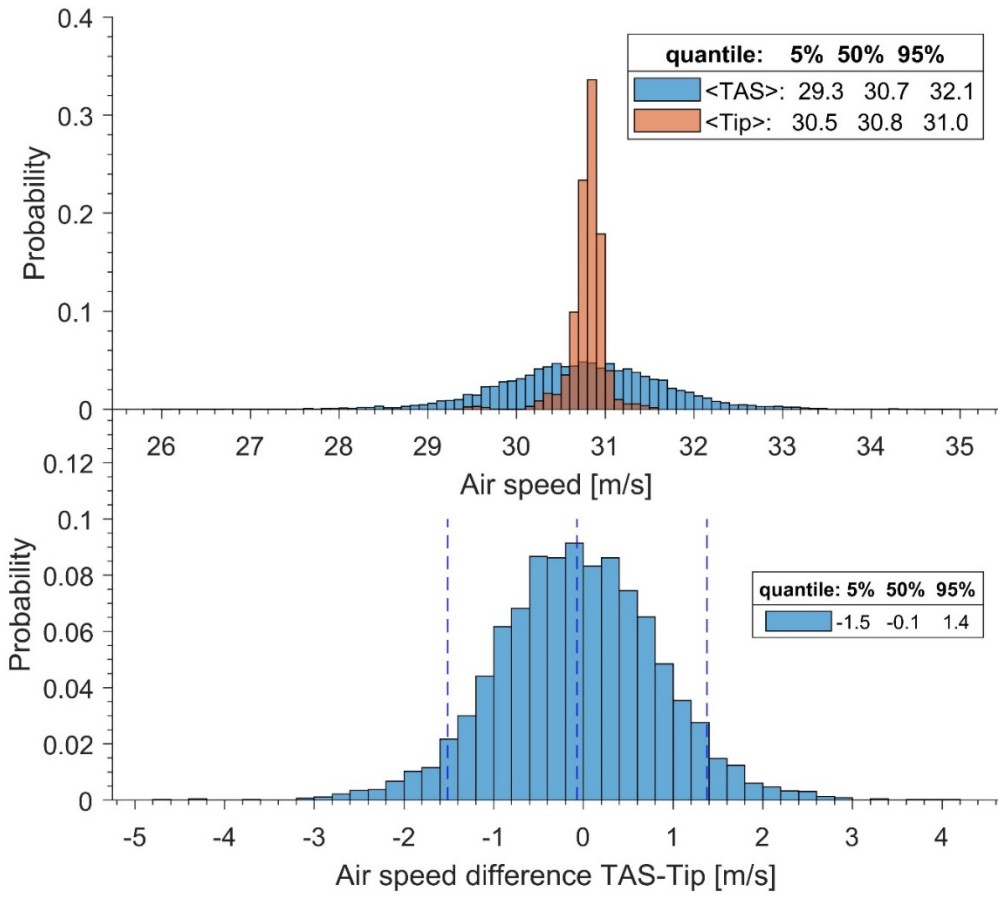


**Figure 8**, The isokinetic inlet performance evaluated using one level flight leg on 11/08/2021

Earth System
Science
Data

Routine calibrations were carried out in the lab to characterize the primary aerosol sensors' performances before and after each deployment, such as the size accuracy determination and evaluation of the aerosol particle detection efficiency(Mei, 2020; Mei

and Goldberger, 2020a, b; Mei et al., 2020 ). In addition, the AGMS was also deployed to the ARM SGP observatory to compare with the standard ARM Observation Station (AOS, #07) between 11/15/2021 and 11/18/2021. As shown in Fig. 9 and Fig. S7, we achieved excellent agreement between the AGMS CPC 3772 and the AOS CPC 3772. Although we did not directly compare the MCPC with the AOS CPC, the above sequenced comparisons ensured the CPC performance consistency among the measurements from the UAS payload, the AGMS, and the AOS site.

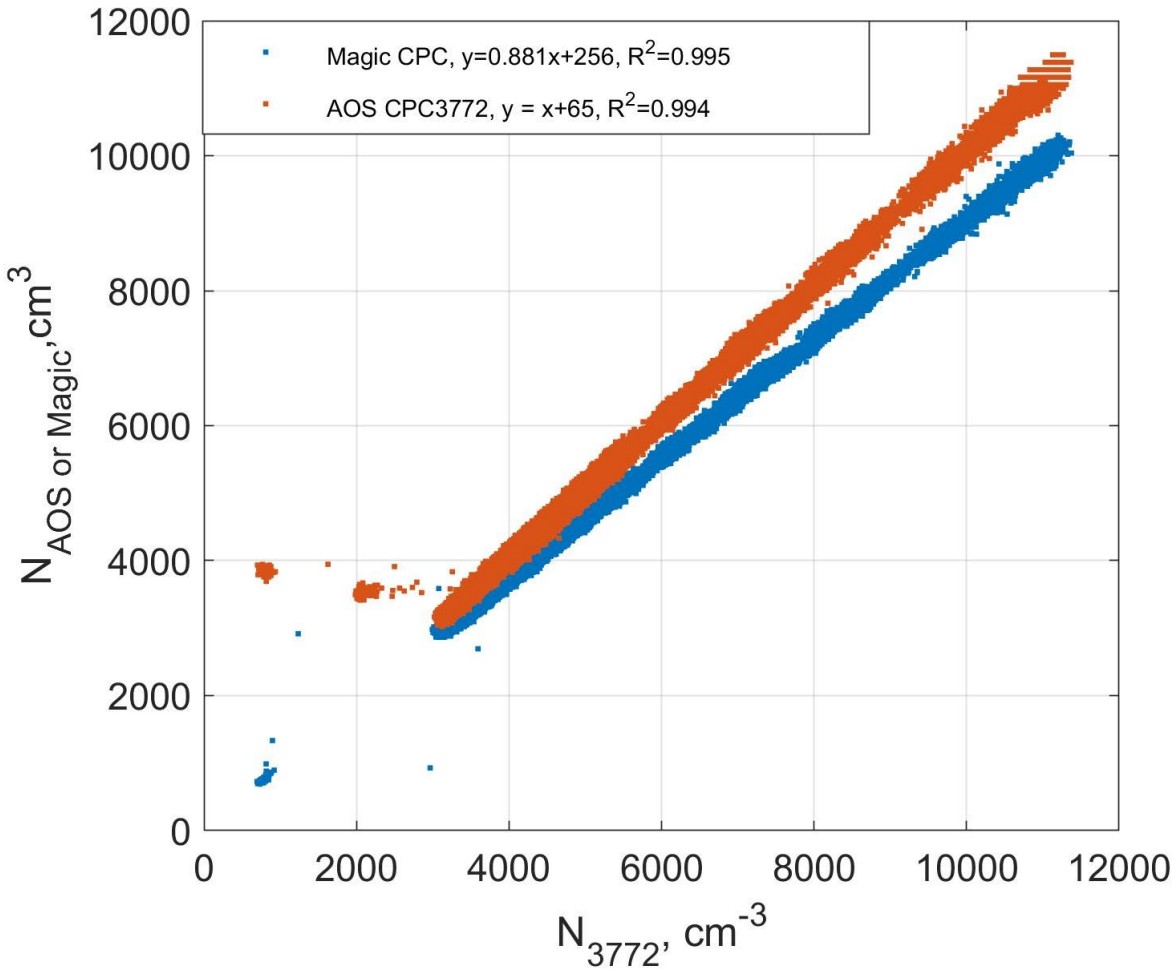


Figure 9. Aerosol total number concentration comparison between the AGMS CPC 3772, the Magic CPC against the AOS CPC 3772 at the ARM SGP observatory between 11/15/2021 and 11/18/2021.

During the UAS flight deployment between 11/6/2021 and 11/15/2021, the UAS aerosol payload was compared against the
AGMS instruments before or after each flight to assess the performance consistency. As shown in Fig. 10 and Fig. S7, the

total number concentration of MCPC was constantly 20% lower than the aerosol concentration of CPC 3772, which indicates

a concentration correction factor should be applied to MCPC data if the UAS data are used to link with the AOS data. In

addition, the water-based CPC is with a 10% uncertainty range compared with the CPC 3772, which is a good payload

candidate.

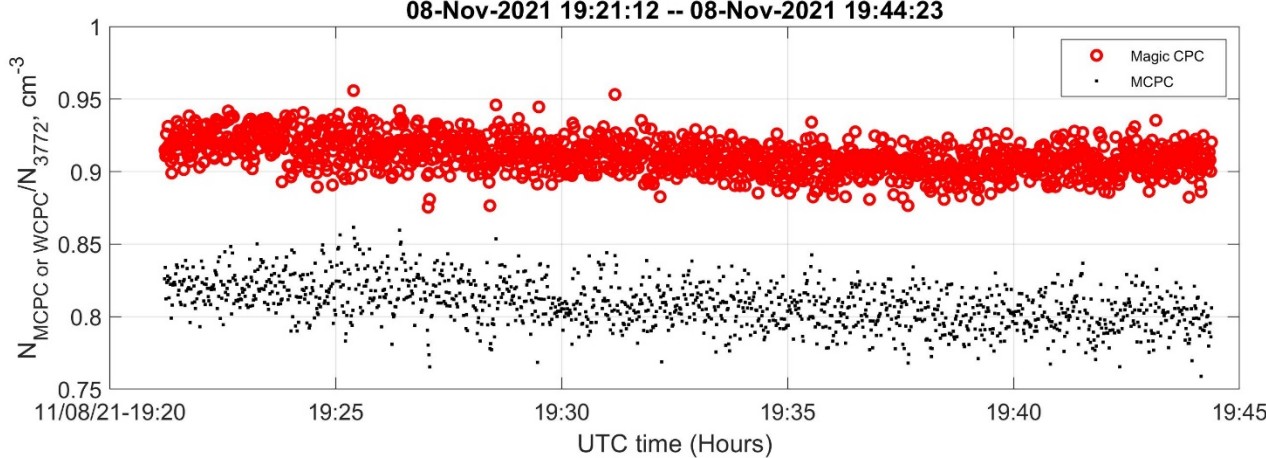


Figure 10. Aerosol total number concentration comparison between the MCPC and the Magic CPC against the AGMS CPC
3772 before the flight on 11/08/2021.

**24-Aug-2021 21:37:00 -- 24-Aug-2021 21:40:55**

Figure 11. Aerosol size distribution comparison among POPS, MOPC, and UHSAS using 200 nm PSL particles.


We also compared the miniaturized size distribution sensors (POPS and MOPC) with their standard-size "sibling", the UHSAS, for 10-30 minutes before each flight using the PSL particles, as shown in Fig. 11. POPS and UHSAS are in better agreement because both instruments were characterized using the PSL particles. The discrepancy between the MOPC and the UHSAS is because the refractive index of the calibration particles used by MOPC differs from POPS and UHSAS. The MOPC used




ammonium sulfate particles to calibrate the aerosol signal in a size range between 180-500 nm. The calibration difference leads
to about a 20% variation in the size determination in MOPC data.

The aerosol absorption measured by STAP agreed well at the ground level with a widely used instrument – a particle soot
absorption photometer (PSAP), as shown in Fig. S8. The in-flight performance of STAP was evaluated in previous studies
(Telg et al., 2017; Düsing et al., 2019; Pikridas et al., 2019). Düsing et al. (2019) pointed out that artifacts in STAP data due
to rapid changes of RH can be corrected using a correction factor of 10.08 Mm$^{-1}$s$^{-1}$ for every 1 % change in RH. Note that no
correction was applied to the flight data in 2021 because there was no significant RH variation during the flight.

Ambient aerosol samples were also collected using a chemical sampler (Mei and Goldberger, 2020a) and analyzed by an
independent research team to confirm the performance of the sensor. A micro-extraction technique was developed to pull out
the aerosol particles collected on the 17 mm polytetrafluoroethylene (PTFE) sample medium for an offline Aerosol Mass
Spectrometer (AMS) analysis. The detailed analysis and results will be discussed in a separate publication currently in
preparation.

### 3.2.3. IRT and multi-spectral imager data

The infrared radiometer deployed on the UAS (IRT, Apogee SI-111-SS) has a 22° half-angle field of view (covering 0.06 km$^2$
surface area from 1 km above the ground) and a response time of 0.6 seconds. The manufacturer calibration has an uncertainty
of 0.2 ℃ when operating between -30-65 ℃. However, the surface temperature recorded by this IRT was influenced by the
ambient temperature at different altitudes. As shown in Fig. 12 (a), the aircraft flew the "ladder" pattern (pattern B) at different
altitudes above the C1 site. With the altitude increase, the measured ambient temperature increased between 900-1200 m, then
decreased between 1200-1500 m, then increased again between 1400-1600 m. This ambient temperature inversion affected the
surface temperature readings obtained by the IRT sensors, shown in Fig. (b). For example, when the ambient temperature
increased between 900 and 1200 m, the IRT measured surface temperature had about 2 -3 ℃ positive bias. Therefore, the IRT
temperature measurements should be performed near the surface or corrected for sensor temperature bias to get an accurate
surface temperature.



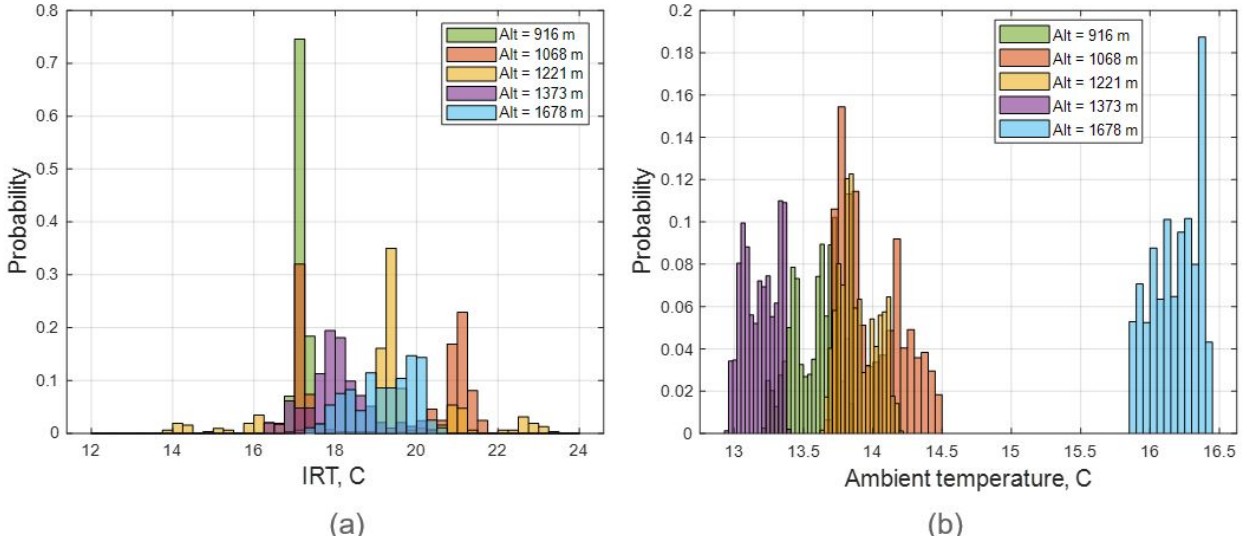

**Figure 12**. Histogram plots of IRT (a) and ambient temperature (b) at each "level" flight leg on 11/09/2021


A multi-spectral camera (Altum) was deployed on the TigerShark for remote surface imagery. The Altum integrates a radiometric thermal camera with five high-resolution narrow bands (475, 560, 668, 717, and 842 nm wavelengths) to produce advanced thermal, multi-spectral and high-resolution imagery, such as surface temperature and green normalized difference vegetation index (GNVDI) images (example in Fig S9). The thermal accuracy of the imager is +/- 5K with thermal sensitivity

of <50 mK. Before every flight, imagery reflectance is calibrated with a reflectance panel and sun sensor. While in flight, the imager recalibrates its thermal camera every 5 minutes or when a 2 °C change in internal temperature occurs. The field of view for the thermal channel of this camera is 48°×39°, which is comparable to the co-located IRT sensor and can be used for cross-checking the IRT sensor in the future. These images were archived separately by the ARM Data Center as user evaluation data and did not follow the data structure described in section 4 as the rest of the ARM UAS/TBS data.

**4.   Data file structure and availability**

All data from the 2021 TBS and UAS deployments are considered as a "routine observation" data stream by the ARM Data management system. The data in this manuscript use Creative Commons License. They were produced following ARM data file standards and archived automatically during and after flight operations through the ARM data ingest process (Mather, 2014; Palanisamy, 2016). Data files are provided to the user community in netCDF format with a naming convention as

follows:

(sss)(inst)(qualifier)(temporal)(Fn).(dl).(yyyymmdd).(hhmmss).nc



where:

(sss) is the three-letter ARM site identifier (e.g., sgp for this data set).

(inst) is the ARM instrument abbreviation (e.g., tbspops, aafmcpc), or the name of an ARM Valu Added Product  (e.g.,
tbsmerge).

(qualifier) is an optional qualifier that distinguishes multiple data from similar instruments but in different platforms.

(temporal) is an optional description of temporal data resolution (e.g., 1s).

(Fn) is the two- or three- character ARM facility designation (e.g., C1, E36).

(dl) is the two-character descriptor of the data level, consisting of one lower-case letter followed by one number (e. g. b1).

(yyyymmdd) and (hhmmss) are the coordinated universal time (UTC) date and time, which indicates the start time of the first
data point measured.

(nc) is the netCDF file extension.

Note that data levels are based on the "level of processing," with the lowest level of data – "00" representing primary raw data
streams collected directly from the instruments. a0 are the raw data converted to netCDF format. With further processing,
including the application of calibration factors and conversion to geophysical units, the data level will be designated to a1-a9.
Furthermore, variables in each data file are checked against the valid range and combined with the position data (latitude,
longitude and altitude), which then will increase the data level to "b". Finally, "c" level data represent value-added data
products providing derived or calculated variables. For example, a netCDF file produced by the TBS SGP deployment from
the POPS that includes quality-controlled data (in geophysical units) collected starting at 17:30:00 UTC on October 8, 2021,
is named as sgptbspopsC1.b1.20211008.173000.nc.

Note that it is not required to have all data levels for each instrument. The ARM data center (ADC) includes sufficient metadata
in each netCDF file to facilitate the user's understanding and interpretation. The time dimension is defined as "unlimited" and
is always the first dimension of a variable using the time dimension. The location variables describe the location of the
measurements, such as latitude, longitude, and altitude. The global attributes include information related to the location of the
platform or the corresponding airbase location (for the aerial platform), the time interval, the calibration procedures if available,
and instrument mentor or principal investigator contact information. Additionally, the individual variables consist of a unique
"long_name" to explain the measured variable with an accurate description of  "units", "missing_value". A "standard_name"
is required if a primary variable and the standard name exist in the Climate Forecast (CF) table.

Files from the UAS/TBS baseline instruments are archived through the ADC (https://www.doi.org/10.5439/1846798) (Mei
and Dexheimer, 2022). The individual DOIs for these datasets are listed in Table 4. Note that a specific directory has been
created for the anonymous reviewer to access the data at https://adc.arm.gov/essd/.

Table 4 DOI for main TBS/UAS datasets

| Data product | Description | DOI/citation |
|---|---|---|



| tbsimet | TBS iMet | 10.5439/1426242 |
|---|---|---|
| tbspops | TBS Portable Optical Particle Spectrometer (POPS) | 10.5439/1827703 |
| tbscpc | CPC flew on the TBS | 10.5439/1827708 |
| aafflitsamp | UAS aerosol filter sampler | 10.5439/1508641 |
| aafh2o | UAS measured concentration of $CO_2$ and $H_2O$ | 10.5439/1507138 |
| aafirt | UAS measured surface temperature | 10.5439/1497848 |
| aafmcpc | UAS measured aerosol total number concentration | 10.5439/1508391 |
| aafmopc | UAS measured aerosol size distribution (0.18 – 10 μm) | 10.5439/1510522 |
| aafstap | UAS measured aerosol light absorption coefficient | 10.5439/1509890 |
| aafmetaims | UAS measured meteorological parameters | 10.5439/1237690 |
| aafnavaims | UAS measured navigation parameters | 10.5439/1238157 |
| aafpopi | UAS measured aerosol size distribution (0.15 – 3 μm) | 10.5439/1778240 |
| aaftrh | UAS measured payload temperature and RH | 10.5439/1506683 |
| Landcover-air | PI processed UAS imagery data | 10.5439/1822938 |

## 5.  Summary

In 2021, 133 TBS flights and seven UAS flights were carried out by staff members from the DOE ARM user facility and the MSU RFRL over the SGP atmospheric observatory. A rich dataset was collected using two platforms (UAS and TBS). TBS data from different seasonal months (in February, May, July and October) provided vertical profiling of aerosol physical properties and atmospheric thermodynamic state at three locations (the central facility and two extended facility EF9 and EF36). In 2022, AAF will continue supporting the TBS data collection with the TBS team from Sandia National Laboratories

at the SGP observatory in February, April and October, and two ARM-approved field campaigns in summer.

UAS data expanded the collective spatial database and provided additional information to assess how measured variables vary spatially and also demonstrated how to compare the mean and turbulent observations derived from SGP remote sensors. After additional training and instrument integration onto the ArcticShark, AAF will carry out more UAS flights over the ARM SGP observatory in July 2022 and pave the path toward routine UAS deployments in conjunction with other ARM facilities.

Paralleling the deployment effort, staff from the AAF, TBS, and ADC teams are also working together to generate merged data products (see example in Table S5) for the user community.

This manuscript provides an overview of the platform and flight patterns used during the data collections and introduces information on data quality control. Those combined observational data can facilitate the current study of the irrigation and urbanization influence on the land-atmosphere-cloud interactions and help improve the understanding of how the naturally-

occurring processes impact on the aerosol-cloud interactions, especially in the boundary layer. We hope that our effort will

encourage broader usage of the ARM data and enhance the collaboration between the ARM user facility and the atmospheric science community.

**Author Contributions.**

FM led the formulation of this paper. FM wrote the draft. FM, DD, MSP, RC, and LAG provided the figures. GdB, JDF, MSP and BS provided editing suggestions to the manuscript. JT, FM, DD, RC, RN, MSP participated in the data collection and preparation of the field campaigns.

**Competing Interests.**

The authors declare that they have no conflict of interest.

**Metadata Information.**

The ARM data center (ADC) includes sufficient metadata in each netCDF file to facilitate the user's understanding and interpretation. Public Data Usage Rights: This work is licensed under the Creative Commons Attribution 4.0 International License. To view a copy of this license, visit https://creativecommons.org/licenses/by/4.0/.

**Acknowledgements.**

This work has been supported by the Office of Biological and Environmental Research (OBER) of the US Department of Energy (DOE) as part of the Atmospheric Radiation Measurement (ARM) and Atmospheric System Research (ASR) Programs. Battelle operates the Pacific Northwest National Laboratory (PNNL) for the DOE under contract DE-A06-76RLO 1830. Gijs de Boer was additionally supported by the US Department of Energy Atmospheric System Research program (**DE-SC0013306**), and the NOAA Physical Sciences Laboratory. We
sincerely appreciate the support from the AAF UAS crew – Peter Carroll, Hardeep S. Mehta, Matt Newburn, and the MSU crew- Austin Wingo, Peter McKinley, Miles Ennis, Connor White, Cavin Skidmore, Nolan Parker, Justin Eskridge, Chase Jackson, Clay Shires, and Randy Welch.
.

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
