# Peer review of "Observational data from uncrewed systems over Southern Great Plains"

_Earth System Science Data, 2022_

## Author Comment (AC1)

The manuscript is well-written and requires only minimal technical edits.

**Response: We sincerely appreciate the comments and suggestions from our reviewer. We address your specific comments below (also in blue). The line number corresponds to the change-tracked version.**

Technical corrections:

line 29: and throughout. Check references with "De Boer" which are listed in the Reference Section as "de Boer".

**Response: changed the reference to "de Boer" in manuscript.**

line 309 state the figure number. Text presently reads "Fig. (b)" which should be Fig. 12 (b).

**Response: corrected it in line 331 to "Fig. 12 (b)".**

line 360: indicate, as stated in the abstract, that access to the data archive requires user registration to the site.

**Response: We added a sentence in line 383. "Note that ADC requires user registration to access the data archive."**

**Citation**: https://doi.org/10.5194/essd-2022-73-RC1

---

## Author Comment (AC2)

General Comments:

The manuscript reports observational data collected using 7 TigerShark unmanned aerial systems (UAS) flights and 133 tethered balloon uncrewed platforms flights. All data has been archived and made freely available. The atmospheric research community may find the data valuable for studying spatial variability of atmospheric and surface parameters. Details of acquisition, collection, and quality control of the datasets are provided with a concise discussion of possible scientific contributions based on these platforms. The manuscript is well-written and only requires attention in the introduction as indicated next. When describing recent work with UAS and tethered balloons in lines 25-31 the manuscript does not sufficiently explain recent efforts implemented to study the Earth System with combined approaches. The manuscript should explain the work in a couple of papers: (a) One from Sensors 2019, 19(8), 1914; DOI: 10.3390/s1908191410.3390/s19081914, which used a balloon-launched unoccupied glider with a suite of sensors to measure atmospheric temperature, pressure, and relative humidity in missions beyond visual line of sight. (b) One from Meteorol. Z. 2009, 18, 141–147; DOI: 10.1127/0941-2948/2009/0363 about a small UAS called SUMO for atmospheric boundary layer measurements. Once this matter is addressed in a minor revision, the revised manuscript should be ready for publication.

**Response: We sincerely appreciate the comments and suggestions from our reviewer. Thank you very much for considering the publication of our manuscript. We added the additional reference in lines 28, 29, and 31. We also revised the paragraph between lin 25 – 40.**

**"Expanding development and deployment of various uncrewed aircraft systems (UAS) result in increasing opportunities for these platforms to provide high-quality atmospheric measurements (Stephens et al., 2000; Hobbs et al., 2002; Villa et al., 2016; de Boer et al., 2020b). Several recent atmospheric science campaigns have provided perspectives on the planetary boundary layer with both UAS (Reuder et al., 2009; Fladeland et al., 2011; Villa et al., 2016; Adkins and Sescu, 2018; Barbieri et al., 2019; Chen et al., 2020; de Boer et al., 2020b; de Boer et al., 2020a) and tethered balloon system (TBS) (Schuyler et al., 2019; de Boer et al., 2018; Creamean et al., 2021). These studies have taken advantage of various scales of UAS platforms, ranging from very small (Schuyler et al., 2019; de Boer et al., 2019) to very large (Intrieri et al., 2014). The above studies developed various cost-efficient sensor systems to probe the Earth system and provided observational data of atmospheric parameters, such as temperature, relative humidity, and wind properties."**